Antimicrobial and cytotoxic effects of marine sponge extracts Agelas clathrodes, Desmapsamma anchorata and Verongula rigida from a Caribbean Island

Piron Julie 1
Betzi Stephane stephane.betzi@inserm.fr 2
Pastour Jessica 1
Restouin Audrey 2
Castellano Rémy 2
Collette Yves 2
Tysklind Niklas 3
Smith-Ravin Juliette 1 4
Priam Fabienne fabienne.priam@univ-antilles.fr fabiennepriam@gmail.com 1 4
1 Groupe de Recherche BIOSPHERES, Université des Antilles, Campus de Schoelcher , Martinique , France
2 Centre de Recherche en Cancérologie de Marseille (CRCM) - Aix-Marseille Université, Inserm, CNRS, Institut Paoli Calmettes , Marseille , France
3 INRAE - UMR 0745 ECOFOG, Campus Agronomique CEDEX , Kourou, Guyane , France
4 Association AREBio Immeuble Bellevue , Fort de France, Martinique , France
Figuerola Blanca
Electronic publication date: 2022 Sep 23
Publication date: 2022
Volume: 10
Electronic Location ID: e13955
Received 2022 Feb 21; Accepted 2022 Aug 5
Copyright: ©2022 Piron et al.
Copyright year: 2022
Copyright holder: Piron et al.
License: This is an open access article distributed under the terms of the Creative Commons Attribution License, which permits unrestricted use, distribution, reproduction and adaptation in any medium and for any purpose provided that it is properly attributed. For attribution, the original author(s), title, publication source (PeerJ) and either DOI or URL of the article must be cited.
License URL: https://creativecommons.org/licenses/by/4.0/

Keywords: Marine sponges, Antimicrobial activity, Cytotoxic activity, Agelas clathrodes, Desmapsamma anchorata, Verongula rigida, Natural products, Martinique, Tumoral cell lines

Funding: Collectivité Térritoriale de Martinique (CTM) Société Anonyme de la Raffinerie des Antilles (SARA) Office De l’Eau (ODE) This work was supported by the “Collectivité Térritoriale de Martinique” (CTM), “Société Anonyme de la Raffinerie des Antilles” (SARA) and “Office De l’Eau” (ODE). The funders had no role in study design, data collection and analysis, decision to publish, or preparation of the manuscript.

==============================
Although marine sponges are known for their antimicrobial, antifungal and cytotoxic activity, very few studies have been carried out on endemic species of Martinique. Martinique is part of the Agoa Sanctuary, a marine protected area that includes the exclusive economic zones (EEZ) of the French Caribbean islands, making it an abundant source of marine species. To highlight the potential of this area for the discovery of marine biomolecules with antipathogenic and antitumor activities, we tested the aqueous and ethanolic extracts of sponge species Agelas clathrodes, Desmapsamma anchorata and Verongula rigida. Five bacterial strains: Bacillus cereus (CIP 78.3), Escherichia coli (CIP 54.127), Pseudomonas aeruginosa (CIP A22), Staphylococcus aureus (CIP 67.8) and Staphylococcus saprophyticus (CIP 76125) were evaluated, as well as four tumor cell lines: breast cancer (MDA-MB231), glioblastoma (RES259) and leukemia (MOLM14 and HL-60). Antimicrobial activity was evaluated using the disc diffusion technique by determining the minimum inhibitory and minimum bactericidal concentrations. Tumor cytotoxic activity was determined in vitro by defining the minimum concentration of extracts that would inhibit cell growth. Ethanolic extracts of Agelas clathrodes were bactericidal for Staphylococcus aureus and Staphylococcus saprophyticus strains, as well as strongly cytotoxic (IC50 < 20 µg/mL) on all cancer cell lines. Verongula rigida also showed strong cytotoxic activity on cell lines but no antimicrobial activity. These results are innovative for this species on these bacterial lines, highlighting the potential of sponge extracts from this area as bioactive compounds sources.

Introduction

Martinique is part of the AGOA Sanctuary, a marine protected area recognized under the Specially Protected Areas and Wildlife (SPAW) protocol that includes the entire exclusive economic zones (EEZ) of the four French Caribbean islands (Saint-Martin, Saint-Barthelemy, Guadeloupe and Martinique), making it one of the “hot spots” of worldwide marine diversity (https://sanctuaire-agoa.fr/editorial/vast-territory) (Fig. 1). Indeed, new sponge species are regularly discovered on the 47,000 km2 EEZ of the island by the active study of its biodiversity (Grenier et al., 2020; Griffiths et al., 2021; Impact Mer, 2008; Impact Mer, 2012; Perez et al., 2017; Perez & Ruiz, 2018). However, these marine organisms are poorly valued (Laville et al., 2009). Therefore, the goal of this study is to improve knowledge of the island’s sponge diversity as a potential source of novel marine natural products with antibacterial activity on drug-resistant pathogens or anticancer activity on common cancer cell lines.

Indeed, bioactive natural products are increasingly sought after, with a specific interest in pharmacology, where they account for about 70% of approved drugs (Cortadellas et al., 2010). In 2017, seven pharmaceuticals derived from marine substances were validated for clinical uses by the Food and Drug Administration (FDA) (Dyshlovoy & Honecker, 2018). Marine sponges are of particular interest because of the abundance of secondary metabolites they produce and their highly diversified chemical nature (El-Amraoui et al., 2010; Mayer et al., 2013; Nweze et al., 2020; Sipkema et al., 2005).

Figure 1 Identification of the sponges.

Map of the sampling location (top) and morphological analysis of each sponge (bottom) showing: (A, D, G) photos after freezing; (B, E, H) micrographs of the skeleton x100; (C, F, I) micrographs of the skeleton x4000. The analysis shows: (C) whorled achantostyles and whorled achantoxes in formation; (F) oxes, isocheles, tripods and spheriaster; (I) thick and short oxes.

The discovery of novel antibiotics from these organisms is one of the main goals of current research. This is due to the increasing resistance of many strains to commercially available antibiotics such as Pseudomonas aeruginosa or Staphylococcus aureus which are ranked among the “priority pathogens” resistant to antibiotics. Marine organisms are a historically rich source of compounds active against these drug-resistant pathogens such as those found in bacteria, fungi, algae or invertebrates (Nweze et al., 2020).

Cancer treatment has also benefited from marine natural products with notable examples like cytarabine, eribulin, and trabectedin. The anti-tumor and cytotoxic activity of sponges have been known and exploited for years with the emergence of chemical derivatives used in cancer treatments such as cytarabine (ARA-C) for the treatment of leukemia or eribulin mesylate as treatment for breast cancer (Dyshlovoy & Honecker, 2018). Despite the existence of these and other treatments, the National Estimates of Cancer Incidence and Mortality in Metropolitan France states that women’s breast cancer incidence increased by 0.6% per year since 2010. During this same period, the incidence rate for acute leukemia increased by an average of 114.5% (Defossez et al., 2019). In light of this, there is an urgent need for new anti-tumor drugs with novel targets and novel modes of action that could be filled by novel marine sources.

This study focused on marine sponges collected on the Caribbean coast of Martinique. We evaluated the activity of aqueous and ethanolic extracts of the three sponge species Agelas clathrodes, Desmapsamma anchorata and Verongula rigida for their antimicrobial and antitumor activity. We highlight the antibiotic and anticancer activity of several sponge extracts after evaluation on five bacterial strains (Bacillus cereus (CIP 78.3), Escherichia coli (CIP 54.127), Pseudomonas aeruginosa (CIP A22), Staphylococcus aureus (CIP 67.8) and Staphylococcus saprophyticus (CIP 76125)), and on four tumor cell lines (breast cancer (MDA-MB231), glioblastoma (RES259) and leukemia (MOLM14 and HL-60)), and are able to link some of our findings to previously reported data.

Material and Methods

Sampling and identification

The samples Agelas clathrodes, Desmapsamma anchorata and Verongula rigida were collected on the “Fond Boucher” site (14°39′21.12″N−61°9′21.22″W) on the north Caribbean side of Martinique (Fig. 1. and Table 1). The samples were collected by dives between 15 and 20 m deep during two campaigns, on October 7, 2017 by Dr. Romain Ferry (2019 Fishing Order No. R02-2019-04-08-004, Martinique Sea Department). The samples were conditioned in individual plastic bags and immediately stored in a cooler for transport to the laboratory of the Université des Antilles (UA). They were then washed with fresh water and directly stored at −20 °C before extraction.

Sponges have been identified on the basis of their external morphology and skeletal composition, in particular the type of spicule (Boury-Esnault & Rützler, 1997; Custódio & RJ, 2007; ImpactMer, 2008; Impact Mer, 2012; Perez et al., 2017; Perez & Ruiz, 2018). The sponge skeletons were studied at the laboratory on UA campus in Martinique, by extraction of the spicules and by longitudinal and transverse sectioning of the tissue. Extractions were carried out by three washes with bleach after freeze-drying 5 mg of sample. Observation was conducted using an optical microscope equipped with a ZEISS camera. The pictures were analyzed with the ZEN2012 software. The size of the spicules was estimated by calculation according to the lens size. The determination of the species was realized with the external anatomic description and observed spicules, as well as the bibliography analysis based on the inventories already published in the area. No comparison was made with similar samples or holotypes. Internal anatomy and cytology were not determined in this study as well.

Method for calculating the size of the spicules: size on the photo/magnification (lens*objective).

Preparation of sponge extracts

The extracts were prepared by maceration in two solvents, distilled water and 100% ethanol. The sponges were dried in a freeze-dryer, then 2 g were cut into small pieces, crushed with a mortar, placed in a 15 mL falcon tube, then 10 mL of solvent were added. After repeated manual turning and swirling, the tubes were placed under agitation for 24 h at room temperature. The extracts were then filtered on standard filter paper. The process was repeated three times. The extracts were then pooled and stored at −20 °C before drying. The ethanolic extracts (E) were dried in a rotary evaporator at 45 °C. one mL of solvent was added to the flask twice and passed through an ultrasonic bath to recover the residues and placed in glass vials. The solvent residues were evaporated in a fume hood and the dry extract was stored at −20 °C before testing. The aqueous extracts (A) were freeze-dried and stored at −20 °C.

Table 1 List of species collected and GPS localization data.

Species	Site	GPS data	
Agelas clathrodes	Fond Boucher	61°9′21.22″W 14°39′21.12″N	
Desmapsamma anchorata	Fond Boucher	61°9′21.22″W 14° 39′21.12″N	
Verongula rigida	Fond Boucher	61°9′26.55″W 14°39′23.67″N	

Antimicrobial activity

Bacterial strains

The five bacterial strains belong to group 1 and 2 of the classification of microorganisms by risk groups (Article R4421-3 of the Decree no. 2008-244 of March 7, 2008-art. (V)). Strains came from the Institut Pasteur Collection (CIP): Paris: Bacillus cereus (CIP 78.3), Staphylococcus saprophyticus (CIP 76125T), Escherichia coli (CIP 54.127) strains are listed in group 1 (non-pathogenic) and Pseudomonas aeruginosa (CIP A22), Staphylococcus aureus (CIP 67.8) strains are listed in group 2 (pathogenic). Inoculums were prepared from strains cultivated in agar nutrient at 37 °C for all strains except for Pseudomonas aeruginosa (CIP A22) incubated at 30 °C.

Antimicrobial assay

Antimicrobial activity was tested on Mueller Hinton agar using the disc diffusion method according to Majali et al. (2015). Pure 18 h culture inoculums on agar nutrient, were seeded at a concentration of 107UCF/mL. Sterile six mm diameter discs were soaked with 20 µL of extract (at 500 µg/mL) and dried for 15min. The discs were then stored at 4 °C before being placed on the seeded agar and the petri dishes were incubated for 24 h at 37 °C for all strains except for Pseudomonas aeruginosa (CIP A22) incubated at 30 °C with agitation. The inhibition diameter was then measured. Tests were performed in triplicates for each species. The standard antibiotic ampicillin (10 µg, lot 7B5479) was used as a positive control for S. saprophyticus, S. aureus and E. coli; chloramphenicol (30 µg, lot 5C5220) was used as a positive control for B. cereus and fosfomycin (50 µg, lot 4L5252) was used as a positive control for P. aeruginosa. Discs impregnated with 20 µL of solvent (H2O and ethanol) were also used as negative controls.

An inhibition diameter greater than nine mm around the disc indicated positive activity according to the analysis of Cita et al. (2017).

Minimal inhibitory concentration (MIC)

MIC was determined by the successive liquid microdilution method according to Majali et al. (2015), modified for ethanolic extract of Agelas clathrodes. Two-in-two dilutions of the crude extract were performed in a 96-well plate. The plates were then incubated 24 h at 37 °C for all strains except for Pseudomonas aeruginosa (CIP A22) incubated at 30 °C. The optical density was measured on a plate reader at 600 nm. The concentration range used was 0.488 µg/mL to 1,000 µg/mL.

Minimal bactericidal concentration (MBC)

MBC was also evaluated for the ethanolic extract of Agelas clathrodes by counting the surviving bacteria in tubes with no visible growth using a method adapted from Majali et al. (2015) and Marinho et al. (2010). A streak of a 10 µL aliquot was seeded onto PCA plates using a calibrated plater, and incubated for 24 h at 37 °C for all strains except for Pseudomonas aeruginosa (CIP A22) incubated at 30 °C. After incubation, colonies were counted for each streak. Only streaks between 30 and 300 colonies were considered. The extract was considered having a bactericidal effect for MBC/MIC = 1 and a bacteriostatic effect for MBC >MIC (Majali et al., 2015).

Cytotoxic evaluation

Cell lines

Four cell lines were selected for the cytotoxic evaluation. A human breast cancer cell line MDA-MB-231, a glioblastoma cell line (RE259) and two leukemia cell lines: MOLM-14 acute myeloid leukemia (more precisely, a MOLM14 luc cell line, expressing the luciferase-GFP gene) and HL-60 acute human promyelocytic cell line. Cell lines came from ATCC (CCL-240) for HL-60; ECACC (cat no. = 92020424) for MDA-MB-231; MOLM-14 GFP/Luc = MOLM-14 were obtained from JE. Sarry and engineered to express luciferase (Stuani et al., 2021) and Res259 (grade II, diffuse astrocytoma) were kindly provided by Chris Jones (The Institute of Cancer Research, Sutton, UK. (Rakotomalala et al., 2021).

Cell lines were cultivated according to the following protocol established by the TrGET facility in the Marseille Cancer Research Center (CRCM).

All cancer cell lines were tested negative for mycoplasma contamination. RES259 cells were grown in MEM medium supplemented with 10% heat-inactivated foetal bovine serum and 1% non-essential amino acids. MDA-MB-231 cells were cultured in RPMI supplemented with 10% FCS, 1% L-Glutamine and 1% sodium pyruvate. MOLM-14 were maintained at a concentration of 0.5 M/mL in MEM alpha medium supplemented with 10% FCS at 37 °C 5% CO2. The HL-60 cell line was grown using the same procedure in IMDM (Iscove’s Modified Dulbecco’s Medium, Gibco 12440053) supplemented with 20% FCS. Adherent cells were seeded in Corning 3903 clear-bottom 96-well plates overnight prior to treatment at 1250 cells for RES259 and 5000 cells for MDA MB231 in 90 µL of their respective medium. MOLM-14 and HL-60 cells were seeded at 10,000 cells per well on the day of testing.

Dilution of the extracts

During cytotoxic tests, only ethanolic extracts were analyzed because contaminations have been noticed for aqueous extracts in the cell cultures. For each sponge sample, two extractions were evaluated (X1 and X2) for a total of 3 performed experiments labelled as #1, #2 or #3. A 10X concentrated cascade dilution of the compounds was performed in medium at 10% constant DMSO concentration and then 10 µL of these dilutions were added to the 90 µL of the wells in triplicate, in order to get concentrations ranging from 500 µg/mL to 0.06 µg/mL (1% final DMSO concentration).

Cytotoxic assay

Experiments were performed as triplicates (except marked otherwise in the result table). Doxorubicin was used as positive control for the MDA-MB231 assay, and Aracytin was used for the three other cell lines. Plates were incubated for 72 h at 37 °C with 5% CO2, then after a 30 min room temperature reset, 50 µl of cell titer Glo (Promega, Madison, WI, USA) were added. Cell lysis was induced for 2 min on an orbital shaker, then after 10 min of incubation at RT to allow the signal to stabilize, the luminescence was measured on a Berthold centro LB960 luminometer. Curves depending on the doses were plotted and the median inhibitory concentration corresponding to the lowest concentration of active compound allowing to inhibit the growth of the cells by 50% in vitro (IC50) was calculated using the GraphPad PRISM software, as well as a confidence interval range (CRI). For IC50 measurements, values were normalized and fitted with GraphPad Prism (least squares regression) using the following equation Y=100/(1+((X/IC50)∧Hillslope)).

Results

Sponges identification

Sponges were identified according to two criteria: their external morphology and the composition of their skeleton.

The first sample was identified as Agelas clathrodes (Table 2). For this sample, the external morphology was massive and the color was bright orange (Fig. 1A). Consistency was hard and oscula were often fused in a comma shape. The skeleton was composed of spongin fibers, achantostyles and achantoxes verticillates megascleres, mostly between 70 µm and 500 µm in size (Fig. 1B). No microscleres were observed (Fig. 1C) (Hooper & VanSoest, 2002a; Hooper & VanSoest, 2002b; VanSoest & Hooper, 2002; Van Soest, 2002).

The second sample was identified as Desmapsamma anchorata (Table 2). External morphology presented a branching of pink-lilac colour with oscules scattered at the surface (Fig. 1D). Consistency was soft and the skeleton consisted of diactinal oxea megascleres (Fig. 1E) of around 80 µm to 100 µm as well as miscrocleres: sigmas, isochelas, spherasters and tripods of less than or equal to 20 µm (Fig. 1F) (Hooper & VanSoest, 2002a; Hooper & VanSoest, 2002b; Van Soest, 2002).

Finally, the last sample was identified as Verongula rigida (Table 2). External morphology was massive with a charcoal black exterior and a sulphur yellow interior (Fig. 1G). Consistency was hard and oscula were located in crevices. They were round, wide and randomly distributed at the surface. The skeleton was mainly composed of spongin fibers (Fig. 1H) and a few short and thick oxeas of about 60–70 µm were observed but were not specific to the sample (Fig. 1I) (Van Soest, 1978; Bergquist & Cook, 1978; Bergquist & Cook, 2002).

Table 2 Identification of sponge species collected.

Sample 1	Class: Demospongiae Sollas, 1885
Sub-class: Heteroscleromorpha Cárdenas, Pérez & Boury-Esnault, 2012
Order: Agelasida Hartman, 1980
Familly: Agelasidae Verrill, 1907
Genus: Agelas Duchassaing & Michelotti, 1864
Agelasclathrodes Schmidt, 1870	
Sample 2	Class: Demospongiae Sollas, 1885
Sub-class: Heterocleromorpha Cárdenas, Pérez & Boury-Esnault, 2012
Order: Poecilosclerida Topsent, 1928
Familly: Desmacididae Schmidt, 1870
Genus: Desmaspamma Burton, 1934
Desmapsamma anchorata Carter, 1882	
Sample 3	Class: Demospongiae Sollas, 1885
Sub-class: Verongimorpha Erpenbeck, Sutcliffe, De Cook, Dietzel, Maldonado, Van Soest, Hooper & Wörheide, 2012
Ordre: Verongiida Bergquist, 1978
Verongula rigida Esper, 1794	

Antimicrobial activity

Antimicrobial assay

In order to evaluate the antibacterial activity of the three extract sponges in water or ethanol, inhibition diameters were obtained by the disc diffusion method on three Gram+ bacterial strains: S. aureus, B. cereus and S. saprophyticus (Fig. 2A); and on two Gram- bacterial strains: E. coli and P. aeruginosa (Fig. 2B).

Figure 2 Antimicrobial screening and antibacterial activity of sponge species by the disc diffusion method on five bacterial strains.

(A) Three Gram+ bacterial strains: S. aureus, B. cereus and S. saprophyticus. T+ (positive control): Ampicillin for S. aureus and S. saprophyticus; Chloramphenicol for B. cereus. (B) Two Gram- bacterial strains: E. coli and P. aeruginosa. T+ (positive control): Ampicillin = for E. coli; fosfomycin for P. aeruginosa. (A) and (B) T1- (negative control 1): H2O only. T2- (negative control 2): EtOH only. S1 H2O: A. clatrodes aqueous extract/S1 EtOH: A. clatrodes ethanolic extract. S2 H2O: D. anchorata aqueous extract/ S2 EtOH: D. anchorata ethanolic extract. S3 H2O: V. rigida aqueous extract/S3 EtOH: V. rigida ethanolic extract. For each panel, visible inhibition discs are marked with yellow arrows.

Different positive controls were used to match bacteria sensitivity (T+). Ampicillin has been used for S. saprophyticus, S. aureus and E. coli; chloramphenicol for B. cereus; and fosfomycin for P. aeruginosa. Water only (T1-) or ethanol only (T2-) were used as negative controls (Figs. 2A and 2B).

Large inhibition diameters were observed for the positive controls and no inhibition diameters were obtained for negative controls as expected (Figs. 2A and 2B).

For the sponge extracts, only the ethanolic extract of Agelas clathrodes showed inhibition diameter for S. aureus and S. saprophyticus (Fig. 2A).

The inhibition diameter was then measured (Table 3) with an inhibition diameter greater than nine mm around the disc indicating positive activity (Cita et al., 2017).

Table 3 shows the antimicrobial activity results obtained on the five bacterial strains. Only the ethanolic extract of Agelas clathrodes shows high specific activity on the two Gram + strains: S. aureus (inhibition diameter: 10.7 mm) and S. saprophyticus (inhibition diameter: 9.5 mm).

MIC and MBC

Only Agelas clathrodes showed inhibition discs, it was therefore possible to measure MIC and MBC values. The MIC determined for S. aureus was 15.62 µg/mL and was higher than the 7.81 µg/mL obtained for S. saprophyticus (Table 4). This result was consistent with the 15.62 µg/mL MBC on S. saprophyticus and the 31.25 µg/mL on S. aureus. We could also note that the MBC/MIC ratios were equal to 2, indicating a bacteriostatic effect for the ethanolic extract of Agelas clathrodes.

Table 3 Inhibition diameter of aqueous and ethanolic extracts of the different sponge species on the five strains (mm).

Extracts sponges and controls	Extract
solvent	Strains	
		Gram +	Gram -	
		S. aureus	B. cereus	S. saprophyticus	E. coli	P. aeruginosa	
Ampicillin (T+)		46 ± 1	–	40 ± 1	26 ± 0	–	
Chloramphenicol (T+)		–	25 ± 0	–	–	–	
Fosfomycin (T+)		–	–	–	–	17.5 ± 0.71	
(H2O) T1-	A	R	R	R	R	R	
(EtOH) T2-	E	R	R	R	R	R	
A. clathrodes	A	R	R	R	R	R	
E	10.66 ± 0.58	R	9.5 ± 0.5	R	R	
D. anchorata	A	R	R	R	R	R	
E	R	R	R	R	R	
V. rigida	A	R	R	R	R	R	
E	R	R	R	R	R	
Notes.

A aqueous extract

E ethanolic extract

R resistant

- not used for this strain

Table 4 MIC and MBC for the ethanolic extract (E) of Agelas clathrodes.

Strain	MIC (µg/mL)	MBC (µg/mL)	MBC/ MIC ratio	
S. saprophyticus	7.81	15.62	2	
S. aureus	15.62	31.25	2	

Cytotoxic activity

Cytotoxic assay

A complete cytotoxicity evaluation of each sponge ethanolic extract was performed on four cancer cell lines: MDA-MB-231 (breast cancer), RE259 (glioblastoma), MOLM-14 (acute myeloid leukemia) and HL-60 acute (human promyelocytic leukemia). Figs. 3 and 4 display these cytotoxic dose response evaluations. Results for synthetic anticancer drugs used as references (doxorubicin and Ara-C) are also indicated. These control experiments were in agreement with previous internal evaluations, thus strengthening the sponge extracts results and conclusions. Several extracts could be linked to strong to moderate cytotoxic activity with cell viability percentages reduced in a dose response manner. All the IC50 extracted from the dose response analysis were kept independent to illustrate the good homogeneity observed across different evaluation experiments and for several extraction campaigns. Table 5 lists all these measured IC50 (per experiment and per extract) as well as the 95% confidence interval range (CIR) generated during the fitting. Overall, the Agelas clathrodes extracts exhibited a strong cytotoxicity on all evaluated cell lines. The Verongula rigida extracts also exhibited strong to moderate cytotoxicity on three cell lines, while the Desmapsamma anchorata extracts were mostly inactive. A detailed individual analysis is presented below for each cell line.

Figure 3 Cytotoxic evaluation of the sponge extracts on the for MDA-MB-231 and RES259 cell lines.

For each sample, three dose response experiments (1st blue; 2nd red; 3rd green) were performed as triplicates. Doxorubicin and Ara-C and were evaluated as positive controls for MDA-MB-231 and RES259 respectively, in two independent experiments.

Figure 4 Cytotoxic evaluation of the sponge extracts on the MOLM-14 and HL60 cell lines.

For each sample, two or three dose response experiments (1st blue; 2nd red; 3rd green) were performed as triplicates. Ara-C was evaluated as positive control in two independent experiments for MOLM14 and one experiment for HL60.

Antiproliferative activity on the human breast cancer line (MDA-MB-231)

The results reported in Table 5, show that the ethanolic extracts of the two species: A. clathrodes and V. rigida were active on cell viability for this cell line. On the other hand, the D. anchorata extracts were inactive. Indeed, the MDA-MB-231 cells were more sensible to A. clathrodes with IC50 values in the µg/mL range (3.42 to 7.91 µg/mL). The V. rigida extracts were also active but with 1 log less potency, exhibiting IC50 values ranging from 18.62 to 64.41 µg/mL.

Antiproliferative activity on the glioblastoma cell line (RES259)

The activity of the extracts on the RES259 glioblastoma cell line was similar to the results observed for the breast cancer cell line. Strong cytotoxicity properties were observed for A. clathrodes and V. rigida with IC50 values lower than 10 µg/mL as shown in Table 5. A. clathrodes is once again the most active extract with IC50 values in the µg/mL range (1.45 to 6.16 µg/mL) followed by V. rigida with again 1 log weaker potency (16.33 to 25.89 µg/mL). In contrast, D. anchorata did not exhibit significant antiproliferative properties.

Antiproliferative activity on the leukemia cell lines (MOLM-14 and HL-60)

The MOLM-14 leukemia cell line was the most sensitive to all sponge extracts, closely followed by HL-60. Indeed, the species A. clathrodes exhibited similar antiproliferative properties on both cell lines in the µg/mL range (0.42 to 1.57 µg/mL on MOLM-14 and 1.82 to 2.5 µg/mL on HL-60 as seen in Table 5). While V. rigida was active on both cell lines with the same activity range, a slight preference was measured for MOLM-14 over HL-60 (3.18 to 11.94 µg/mL for MOLM-14 versus 11.03 to 36.19 µg/mL for HL-60). While one experiment (#3, X2) could generate a dose response curve for D. anchorata with an IC50 below 100 µg/mL (47.72 µg/mL), this result could not be confirmed for the two other evaluations confirming the overall lack of activity of this species extracts during all the cytotoxicity evaluation and on all cell lines.

Discussion

Antimicrobial activity

The results obtained highlight the effects of Agelas clathrodes whose ethanolic extract is the only sample presenting a specific activity on three strains of staphylococcus.

Table 5 Cytotoxic evaluation results.

Half maximal inhibitory concentration (IC50) and confidence interval range (CRI) measured for the ethanolic extracts (E) on different cancer cell lines (MDA-MB231 (breast cancer), RES259 (glioblastoma), MOLM-14 (leukemia) and HL-60 (leukemia)). For each sponge sample, two extractions were evaluated (X1 and X2) for a total of 3 experiments labelled as #1, #2 or #3. All values are listed as µg/ml concentrations and ‘ND’ is specified for non-determined values and ‘NC’ for not converged values during the fitting process. Doxorubicin and Ara-C are used as positive control anticancer agents.

		MDA-MB-231	RES259	MOLM-14	HL-60	
		IC50**	95% CIR**	IC50**	95% CIR***	IC50**	95% CIR***	IC50**	95% CIR***	
Doxorubicin	#1	0.20	0.14 to 0.29	ND	-	ND	-	ND	-	
#2	0.18	0.13 to 0.23	ND	-	ND	-	ND	-	
Ara-C	#1	ND	-	0.07*	0.05 to 0.08	0.085*	0.06 to 0.11	ND	-	
#2	ND	-	0.08	0.07 to 0.10	0.222	0.15 to 0.25	0.12	0.10 to 0.15	
Agelas clathrodes	#1 (X1)	3.42	2.38 to 4.91	1.45	0.91 to 2.07	0.41	0.34 to 0.51	ND	-	
#2 (X1)	7.91	2.02 to 30.99	3.52	2.64 to 4.66	1.57	1.30 to 1.85	1.82	1.58 to 2.18	
#3 (X2)	7.71	NC	6.16	NC	1.21	1.06 to 1.37	2.5	2.20 to 2.85	
Desmapsamma anchorata	#1 (X1)	>100	-	>100*	-	>100*	-	ND	-	
#2 (X1)	>100	-	>100	-	>100	-	>100	-	
#3 (X2)	>100	-	97.15	NC	47.72	40.04 to 56.95	>100	-	
Verongula rigida	#1 (X1)	18.62	16.77 to 20.67	16.33	NC	3.83	2.74 to 5.20	ND	-	
#2 (X1)	30.68	26.06 to 36.11	11.83	NC	3.18	2.18 to 4.56	11.03	8.25 to 14.50	
#3 (X2)	64.41	54.41 to 76.25	25.89	NC	11.94	6.40 to 20.65	36.19	NC	
Notes.

* Duplicate instead of triplicate.

** Half maximal inhibitory concentration (µg/mL).

*** CIR: confidence interval range (µg/mL).

In similar tests carried out with the Agelas sventres species, such strain-specific activity on staphylococcus had not been observed. In fact, an activity had been observed on E. coli (CIP 54.127) for methanolic extracts and in n-hexane. Similar results were obtained on S. aureus (CIP 67.8) with chloroform and hexane extracts, and finally on C. albicans (ATCC 10231) with chloroform extracts (Galeano & Martínez, 2007). The specificity observed with A. clathrodes probably indicates a difference in active biomolecule composition due to either the species or the nature of the solvent. Indeed, with the aqueous extract having no effect, we can conclude that the nature of the solvent plays a crucial part in the extraction of active molecules. A test performed using a crude methanolic extract of Agelas sp. showed a significant effect on the same S. aureus strain (CIP 67.8), but again these were not specific to staphylococcus (Balansa et al., 2020). Thus, the alcohol extracts appear to have a greater effect on staphylococci than on other strains. Alcoholic solvents would probably allow a better extraction of alkaloid type biomolecules. Indeed, the alkaloids present in marine sponges are known for their antibacterial properties, particularly on S. aureus. For example, bromo-pyrrole alkaloids extracted from the species Agelas dispar, are known for their moderate antimicrobial activity on Gram +: B. subtilis and S. aureus (Chairman, AR & Ramesh, 2012). This suggests the potential presence of similar biomolecules in the A. clathrodes extract or a biomolecule with similar effect. It could also be a combination of biomolecules. The lack of effect on these same strains with the aqueous extract confirms the necessity of an alcoholic solvent for this type of biomolecules. Similarly, in Agelas dilatata, pyrrole-imidazoles were tested on two pathogenic strains of P. aeruginosa (CIP A22 and PAO1) and showed a moderate to strong activity. More specifically, oroidin 1 (pryrrole-imidazole which was first isolated in Agelas oroides in 1971) has also showed a moderate activity against laboratory strains of P. aeruginosa (PA01 and PA14) (Melander et al., 2016). Bromoageliferin is an isolated molecule that showed significant activity, specifically on the P. aeruginosa strain CIP A22 (Pech-Puch et al., 2020, Pech-Puch et al., 2020b). This could indicate the absence of this molecule in A. clathrodes.

Concerning the species V. rigida, this is the first study to analyze the antibacterial activity of this extract on non-marine bacterial strains (Newbold et al., 1999). Indeed, this species is known for its antibacterial effect on some sponge pathogenic strains, in particular Bacillus sp. and Vibrio alginolyticus, but no work has been done on the evaluation of their possible activity on non-marine strains. In addition, antiparasitic (Putra, Hadi & Murniasih, 2016a; Bianco et al., 2015) or antidepressant (Zhang, 2020) effects have also been described. Also, based on our results, we find that this species does not possess activity on non-marine bacterial strains.

Regarding D. anchorata species, we confirmed its lack of activity against Gram- bacteria, including E. coli and P. aeruginosa (Bianco et al., 2015). Indeed, it is becoming increasingly difficult to find antibacterial molecules effective against Gram- bacteria, as confirmed by our results. Moreover, antimicrobial activity tests performed with D. anchorata extracts revealed no effect on S. aureus, a Gram+. This study confirmed the inactivity of the extract on the strains selected for this study but it may be active on other bacterial strains not used here.

The determination of MIC and MBC could only be performed on A. clathrodes showing bioactivity on the selected bacterial strains. The extracts were more potent for S. saprophyticus than S. aureus. Given the quality of the extractions performed and the quantity of extracts obtained after drying, a higher MIC for S. aureus does not mean an absence of active biomolecules but maybe a lower concentration. Indeed, a higher MIC has already been obtained for an alcoholic extract on E. coli for the S. massa sponge (Putra, Hadi & Murniasih, 2016a). Similarly for the aqueous extract, the number of biomolecules may be too low to observe a significant effect on the bacteria.

The bacteriostatic effect of this extract may, again, indicate the presence of biomolecules different from those known to exist in the other studied Agelas genus species.

Moreover, Agelas genus seems to have a specific type of alkaloid biomolecules or a specific combination of them, hence the absence of effect for the species D. anchorata and V. rigida.

Cytotoxic activity

The cytotoxicity evaluation highlighted that the Agelas clathrodes species has a strong cytotoxic activity in addition to antimicrobial properties as previously demonstrated. Agelas clathrodes species is also the one with the highest activity on all cell lines followed by Verongula rigida. Their order of activity for both these sponges is the same, namely, a stronger activity on the leukemia cell lines (MOLM-14 and HL-60) followed by the activity on the RES259 glioblastoma cells and finally on the MDA-MB231 breast cancer lineage.

The sponges of the Agelas genus are known for their cytotoxic effect on many cancerous cells associated with several extracted molecules such as agelasins, agelasidins and agelines. The Agelasphins extracted from the A. mauritianus species exhibit, notably, antitumor and immunostimulatory effects (Natori et al., 1994). Agelasine B extracted from A. clathrodes species has demonstrated effects on the MCF-7 line of human breast cancer cells (Dipolo & Suarez, 2012). Moderate effects of A. clathrodes extracts were also observed on the MDA-MB-435 cell line (Custódio & RJ, 2007).

In our study, we observed similar effects on a different breast cancer cell line. Nevertheless, working with a crude extract, we could not establish here if it was an effect of agelasine B or if it was due to a different molecule or set of molecules. Indeed, other tests conducted on several species of the Agelas genus (A. citrina, A. clathrodes, A. dilatata and A. sceptrum) on the same MCF-7 cell line showed more negligible effects, except for A. citrina which stood out with a percentage of inhibition of 100% at 30 and 15 µg/mL (Pech-Puch et al., 2020, Pech-Puch et al., 2020b). Similar results for extracts of the Agelas clathrodes species were observed in the MDA-MB-435 cancer cell line, as well as for Agelas sp. in the same study (Custódio & RJ, 2007). Furthermore, a moderate activity had been found for A. clathrodes in the HL-60 cell line (IC50: 48.51 µg/mL) as well as in a glioblastoma cell line SF-295 (IC50: 62.36 µg/mL) (Custódio & RJ, 2007). The stronger activity in our evaluation campaign could be explained by a synergy of active biomolecules.

The ethanolic crude extract of Verongula rigida showed a high cytotoxic response in this study. This activity is specific to leukemic (MOLM-14 and HL-60), glioblastoma (RES 259) and breast cancer (MDA-MB231) cell lines. Although no specific studies have been performed on these cells, the published results of (Galeano et al., 2011) on the cytotoxicity of V. rigida on U937 cells of the human myeloid linage, could sustain our observations. Indeed, Galeano showed that this property would be due to aeroplysinin-1, a protein tyrosine kinase inhibitor found in V. rigida extracts. These active biomolecules will have to be purified from the extracts and evaluated to confirm our preliminary results.

As for the antimicrobial and cytotoxic activities of crude ethanolic extract of Desmapsamma anchorata, we validated a lack of activity on our cell lines for the first time, but this was also shown on other cell lines in previous studies (Lhullier et al., 2019; Marques et al., 2016). This was presumably due to the presence of relatively low amount of active biomolecules.

Conclusion

Taken together, the results of this study suggest that crude extracts of marine sponges from Martinique have a potential antimicrobial and cytotoxic effect. Two sponge species stood out: Agelas clathrodes for both antimicrobial and antitumor properties, and Verongula rigida for its antitumor properties. The existence of such active compounds in this species extracts would respond to a need for natural antibacterial and antitumor molecules, with limited side effects or drug resistances. Hypotheses have also emerged concerning the biomolecular composition of Agelas clathrodes and Verongula rigida such as the presence of alkaloids and aeroplysin-1, or biomolecules with similar effects. Moreover, having identified these species in Martinique constitutes a key element for the valorization of the marine biodiversity of the island within the Caribbean area. As this study is a preliminary work never done before in the French West Indies, especially in Martinique, it would be interesting to widen the spectrum of study to other species of the region and to carry out complementary chemical analyses in order to isolate and confirm the bioactive candidate or to identify other biomolecules of interest.

Supplemental Information

Supplemental Information 1 Antibacterial Assay Raw data

Raw data for antibacterial assay: (A) Petri dishes photos. (B) MIC and MBC research. (C) Phot reading plate. (D) Count% viability. (E) Summury tables.

Click here for additional data file.

Supplemental Information 2 Cytotoxicity Assay Raw data

Cytotoxicity analysis of our sponge species on three cancer cell lines by luminescence measurement: MDA-MB-231, MOLM-14, HL-60 and RES259. Luminescence measurements are expressed as RLU (Relative Light Unit).

Click here for additional data file.

The authors thank Dr. Ferry for his involvement in the sampling and identification of sponges. They also thank Dr. Stéphanie Morin, for their help in the realization of antibiograms. The authors also thank Dr. Thomas Miller for helping rewriting the manuscript. This work was part of the PO FEDER project No. MQ0023978.

Additional Information and Declarations

Competing Interests

Author Contributions

Field Study Permissions

Data Availability

The authors declare there are no competing interests.

Julie Piron performed the experiments, analyzed the data, prepared figures and/or tables, authored or reviewed drafts of the article, and approved the final draft.

Stephane Betzi conceived and designed the experiments, analyzed the data, prepared figures and/or tables, authored or reviewed drafts of the article, and approved the final draft.

Jessica Pastour performed the experiments, analyzed the data, prepared figures and/or tables, and approved the final draft.

Audrey Restouin performed the experiments, prepared figures and/or tables, and approved the final draft.

Rémy Castellano conceived and designed the experiments, prepared figures and/or tables, and approved the final draft.

Yves Collette conceived and designed the experiments, prepared figures and/or tables, and approved the final draft.

Niklas Tysklind conceived and designed the experiments, authored or reviewed drafts of the article, and approved the final draft.

Juliette Smith-Ravin conceived and designed the experiments, analyzed the data, authored or reviewed drafts of the article, and approved the final draft.

Fabienne Priam conceived and designed the experiments, analyzed the data, prepared figures and/or tables, authored or reviewed drafts of the article, and approved the final draft.

The following information was supplied relating to field study approvals (i.e., approving body and any reference numbers):

The sponge samples were taken in Fins Mask Snorkel (FMS) in 2017 and were authorized in FMS before the 2019 recreational fishing decree. We had also obtained the verbal agreement of Mrs. Sabrina MUNIER, in charge of marine and coastal biodiversity, marine environment referent - DEAL, Martinique.

The following information was supplied regarding data availability:

The raw data for antibacterial assay and cytotoxicity assay are available in the Supplementary Files.

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
