# Peer review of "Antimicrobial and cytotoxic effects of marine sponge extracts Agelas clathrodes, Desmapsamma anchorata and Verongula rigida from a Caribbean Island"

_PeerJ, doi:10.7717/peerj.13955_

## Round 0.1 · original submission · Major Revisions

Dear Dr. Priam,

Your manuscript has been evaluated by three peer reviewers, and the reviewer comments are appended below.

The reviewers have highlighted a few concerns. All reviewers think the manuscript requires in-depth editing and restructuring and I agree with them. Reviewer #1 thinks the topic is interesting although she/he has raised additional points that need to be addressed (taxonomy and data analysis ). Reviewer #2 thinks the manuscript needs extensive revision including a more detailed information in the Methods. Reviewer #3 also thinks the manuscript needs improvement in terms of context of taxonomy and discussion. While two reviewers also comment the novelty of this study is very limited, "novelty" is not one of PeerJ's editorial criteria.

Based on the referees' recommendations, I arrive at this decision: The manuscript does merit publication in PeerJ but it is not acceptable in its current form and needs a major revision based on the reviews and my general editorial comments below. I therefore invite you to resubmit a revised version.

Please carefully consider the comments of the reviewers and provide a point-by-point response which clearly defines the changes made. Note that reviewer #1 has also provided his/her advice in an annotated pdf file.

Thank you for your patience with the evaluation process and for choosing PeerJ.

I look forward to receiving your revised manuscript.

Yours sincerely,


Blanca Figuerola

Academic Editor, PeerJ
* * *
Editor's general remarks

Please have the language in your manuscript checked by a colleague or consider seeking out academic editorial services.

I also suggest adding the author(s) of the scientific name and the year it was published when a species' name is cited in the manuscript for the first time as recommended by the International Code of Zoological Nomenclature (ICZN).

Reviewer 1 ·

Basic reporting

The English language should be improved to ensure that an international audience can clearly understand your text. Some examples where the language could be improved include lines 23, 77, 121, 128, 233-237, the current phrasing makes comprehension difficult. Also, the use of tenses in the text needs improving. My suggestion is that you ask a colleague who is proficient in English and familiar with the subject matter to review your manuscript, or contact a professional editing service.
Your second important issue is the taxonomy of the sponges identified based on their morphological and spicule characteristics. Although several literature were cited for the spicule preparation methods, there is no single literature on Agelas and the other two species. Please provide the appropriate literature to support the claim made for each species.

Experimental design

Your introduction needs more detailed explanation and restructuring. I suggest that you improve the description at lines 59-81 to provide more justification for your study for example by explaining about the current resistant issues both existing antibacterial and anticancer drugs to expand upon the knowledge gap being filled. Also, for lines 82-87, it can be improved by adding information that sponges from specific or remote areas may contribute to the discovery of novel molecules because of their endemicity in addition to giving understanding of the biodiversity particular areas using an integrative approach such as taxonomy, chemistry, microbiology (see Gordo et al., 2019, doi:10.3390/md17060319).
Methods especially for sponge morphological identification should be described in more detail for others to replicate.

Validity of the findings

It is great that you provide the raw data on sponge spicules along with their size and types. However, you need to provide more supplemental data to be useful to future readers. The data analysis should be improved in the following ways. Because you mentioned that the picture of the spicules were obtained by a light microscope, it raised a question of how did you measure the sizes of spicules that you claim on the text because not all light microscope equipped with measuring equipment.
In addition, your most important issue is the antibacterial and anticancer activities of Agelas clathrodes and Veronginda rigida. They are discussed in the paper but need a better restructuring. It can be achieved for example by first explaining about the antibacterial activity of A. clathrodes then V. rigida followed by the explanation about their MIC and MBC. Regarding the inactivity of D. achoroata, it can be simply mentioned as active against the tested bacteria without having to mention it as a negative control in this study to confirm the earlier study or defending its low amount for the absence of bioactivity. Presumably the extract would be active against other types of bacteria or cancer cell lines but not against the ones examined in this study. Perhaps, the absence of activity against negative bacteria in this study should be mentioned instead, confirming the difficulty in discovering antibacterial for Gram negative bacteria. As for the presence of bromopyroles in A. clathrodes and aeroplysion in V. rigida, they need further proofs (i.e. molecular weight or even better NMR data that can be compared to the reported ones) because at the moment, they are too speculative.

Additional comments

I appreciate the authors for their effort in pioneering the investigation of rich yet relatively untapped marine sponges of the Martinique island as a new source of antibacterial and anticancer agents. In addition, the authors have morphologically identified all the specimens, providing the pictures of both sponges and their spicules, one of the most challenging marine invertebrates to be identified. Nevertheless, the manuscript needs improvement in terms of context, additional experiment for the presence of the claimed alkaloids and grammatical aspects before Acceptance.

Annotated reviews are not available for download in order to protect the identity of reviewers who chose to remain anonymous.

·

Basic reporting

The article describing the anti-microbial and cytotoxic effects of crude extracts of sponges. The article over all doesn’t contains any novelty, only simple Disc diffusion assay on ethanolic extracts and cytotoxic activity is not so satisfactory. This kind of results are not enough to be considered for PeerJ, the article really needs extensive revision. From head till tail there are bundle of mistake among which I discussed downside.


1. More data, more satisfactory results needed to be added for this work.
2. From abstract till end all the name of microbes should be written in italic. Like in abstract the name of staphylococcus should be like staphylococcus. Kindly carefully revise the whole paper.
3. The IC50 should be written like IC50. 50 should be written in lower superscript. Kindly carefully revise the whole paper.
4. The English language is very poor in the manuscript and some sentences needed to be totally modified. It was very confusing for me to read some sentnces again and again to understand. E.g these sentences of abstract. “Specific antimicrobial activity of ethanolic extract of Agelas clathrodes on Staphylococcus aureus and Staphylococcus saprophyticus strains and a strong cytotoxic activity (IC50< 20 μg/mL)on all cancer cell lines have been observed.” I don’t know what authors want to say?. Kindly check ur paper from a fluent English language speaker for all kind of language errors, mistakes, and typo and grammatical errors.
5. The introduction is very poorly described. It needed to be improve.
6. There are two different kind of fonts from line 91-97 and from 98-102. Kindly arrange it in a proper manner.
7. From where the bacterial strains have been achieved? ATCC or hospital or from where? Mention its origin and place.
8. NO need to mention citations in conclusion section. Conclusion is very badly explained.

Experimental design

The experiments are poorly organized. Not well discussed.

Validity of the findings

The results do not contain any novelty. Simple diffusion assay and cytotoxic assay of crude extracts with these results are not enough for PeerJ journal.

Reviewer 3 ·

Basic reporting

1. English title needs to be improved. The word activity is not appropriate for the two screenings performed.
2. Abstract writing needs to be improved and written in proper English. The abstract must clearly explain the background, objectives, research methods, results and conclusions of the study.
3. In the description of the background, it is explained that the need for new antibiotics is very urgent because of the presence of resistant bacteria. It is recommended that the test bacteria used in screening for antibacterial activity are several resistant and sensitive bacteria.
4. Sentences 85 - 87 are not in line with the research background.

Experimental design

1. In general, this study is in accordance with the journal Aims and Scope. Research questions are well defined, relevant & meaningful. Investigations are conducted to the highest technical & ethical standards.
Methods explained with enough detail & information to replicate.
2. Table 2 shows the results of the antibacterial activity test. There is only one active extract so it does not need to be written in tabular form.
3. the Line 139 needs to describe an accurate temperature used for Bacterial incubation
4. Sentence in 138 mentions pure extract..needs to be clarified again because the extract is not pure.
5. In general, the discussion did not analyze in depth the results of the bioactivity test. Authors should carry out a literature study regarding the secondary metabolite content and bioactivity of each previously reported sponge.

Validity of the findings

1. The novelty of this study is very limited.
2. Marine sponge species from Martinique have not been studied much, but why the author only reports 3 species. The number of sponges studied should be in larger numbers.
3. Sponge identification is carried out based on the morphology of the spicules, it should be discussed in more detail by comparing them with available literature or herbarium specimens.

---

## Round 0.2 · Minor Revisions

The authors have improved their review according to the referees' comments although some minor revisions are still required (see reviewer comments). Note that reviewer 1 has also provided their advice in an annotated pdf file.

Reviewer 1 ·

Basic reporting

It is great that the authors already addressed issues raised earlier about the introduction of the manuscript, the method of measuring spicules, and primary references for sponges species identification. However, the authors failed to do so for tenses and grammatical issues especially the inconsistency in the use of tenses and a few grammatical errors which I annotated in the previous review and again in this second review. Please take time and have a look more closely at the annotated document and respond accordingly.

Experimental design

The method for measuring the spicules of three sponges was mentioned in the revised manuscript by the authors. So no comment.

Validity of the findings

As mentioned in the early review, the authors have not addressed the issues about additional information to support their claim on the presence of bioactive molecules such as aeroplysin-1. Without solid proof (e.g. molecular weight LCMS analysis, proton NMR spectrum or simple alkaloid test, etc), the claim of the presence of aeroplysin-1, based only on bioactivity inferred only from different species of sponge, has no solid ground. So, it would be better to support the claimed molecules even with one of the simple tests for instance the presence of an alkaloid if such molecule is being claimed.

Additional comments

The authors also still need to rewrite two paragraphs (lines 400 to 410) to make it more readable to international audiences. Also, the final phrase in the manuscript (lines 425-426) needs rewriting because the phrase, to identify the known but especially unknown active molecules is rarely used. Think about repurposing the known compounds for new bioactivities or identifying new uses/bioactivity for the known compounds.

Annotated reviews are not available for download in order to protect the identity of reviewers who chose to remain anonymous.

Reviewer 3 ·

Basic reporting

no comment

Experimental design

no comment

Validity of the findings

Figure 2 is not clear or too small, so the zone of inhibition on the disc is not clearly visible. The quality and size of the photos need to be improved.

---

## Round 0.3 · accepted · Accept

The authors did a great job addressing the referee’s comments and improving the manuscript.